# scBiGNN: Bilevel Graph Representation Learning for Cell Type Classification from Single-cell RNA Sequencing Data

**Rui Yang**[1,2]**, Wenrui Dai**[1]**, Chenglin Li**[1]**, Junni Zou**[1]**, Dapeng Wu**[2]**, Hongkai Xiong**[1]
[1]Shanghai Jiao Tong University,   [2]City University of Hong Kong
{rui_yang, daiwenrui, xionghongkai}@sjtu.edu.cn, dapengwu@cityu.edu.hk

## Abstract

Single-cell RNA sequencing (scRNA-seq) technology provides high-throughput gene expression data to study the cellular heterogeneity and dynamics of complex organisms. Graph neural networks (GNNs) have been widely used for automatic cell type classification, which is a fundamental problem to solve in scRNA-seq analysis. However, existing methods do not sufficiently exploit both gene-gene and cell-cell relationships, and thus the true potential of GNNs is not realized. In this work, we propose a bilevel graph representation learning method, named scBiGNN, to simultaneously mine the relationships at both gene and cell levels for more accurate single-cell classification. Specifically, scBiGNN comprises two GNN modules to identify cell types. A gene-level GNN is established to adaptively learn gene-gene interactions and cell representations via the self-attention mechanism, and a cell-level GNN builds on the cell-cell graph that is constructed from the cell representations generated by the gene-level GNN. To tackle the scalability issue for processing a large number of cells, scBiGNN adopts an Expectation Maximization (EM) framework in which the two modules are alternately trained via the E-step and M-step to learn from each other. Through this interaction, the gene- and cell-level structural information is integrated to gradually enhance the classification performance of both GNN modules. Experiments on benchmark datasets demonstrate that our scBiGNN outperforms a variety of existing methods for cell type classification from scRNA-seq data.

## 1 Introduction

Single-cell RNA sequencing (scRNA-seq), which allows the measurement of gene expression at the resolution of individual cells, has revolutionized transcriptomic analysis and greatly improved the understanding of biomedical science over the past decade. Accurate cell type annotation is essential for elucidating cellular states and dynamics in scRNA-seq data mining [1] and contributes significantly to a broad range of downstream analyses, such as cancer biology [16] and drug development [22]. In early studies, the cluster-then-annotate paradigm is commonly used to categorize cell types [8, 17]. These methods rely on manually selected marker genes and require prior knowledge of cell types, which is highly subjective and prone to errors due to the unknown quality of clustering results. As copious annotated scRNA-seq data become publicly available, many classification methods have been developed for automatic cell type labeling.

Existing classification methods can be grouped into three categories: 1) traditional machine learning algorithms, 2) similarity-based measurement and 3) deep learning models. The first category of methods applies classic machine learning approaches to scRNA-seq data analysis, such as random forest, linear discriminant analysis and support vector machine. Typical works include CasTLe [14],

NeurIPS 2023 AI for Science Workshop.

scPred [2] and scID [4]. These methods commonly have limited model capacity and do not scale well to large datasets. The second category is based on some similarity criteria to measure the correlation between the unlabeled cells and the reference dataset. For example, scmap [12] calculates the similarity between the query cell and the median gene expression values for each cell type in the reference dataset, while SingleR [3] computes Spearman correlation between cells using variable genes. Similarity-based methods are heavily affected by the batch effect caused by variant experimental conditions [9], and the adopted similarity measurement may not be suitable for the high-dimensional and sparse scRNA-seq data [30]. To enable scalable and robust cell type identification, deep learning models have been increasingly explored and achieved superior performance. For instance, ACTINN [15] first uses fully connected neural networks for cell type identification. scCapsNet [27] employs capsule networks [20] for interpretable gene feature selection in cell representation learning, which improves the reliability of deep learning models. Furthermore, graph neural networks (GNNs) have emerged as widely used tools for cell type annotation [28, 32, 13, 30, 26], as they can leverage the intrinsic biological networks (e.g., gene-gene interaction network) within gene expression profiles for more expressive representation learning on scRNA-seq data.

However, several problems still impede GNN-based classification methods from reaching their full potential. Firstly, although it is generally acknowledged that gene-gene and cell-cell relationships are both beneficial to scRNA-seq analyses [13, 30], none of the existing models consider these two kinds of structural information simultaneously. To be concrete, gene-level GNNs classify cells from the local biological perspective of each cell's own gene expression profile, which leverages informative gene interactions to improve cell representations but relationships between different cells are ignored. Cell-level GNNs consider the connectivity between cells from the global view of the whole database but cannot well capture the fine-grained gene-level biological contexts. Secondly, most existing works rely on predetermined biological networks for graph representation learning. For example, sigGCN [28] utilizes the STRING database [21] to construct gene-gene interaction network, while Li et al. [13] consider to precompute the cell-cell graph by $k$-nearest neighbors analysis based on the gene expression data. Nevertheless, it is hard to evaluate whether these graphs constructed by task-irrelevant knowledge are optimal for cell type classification. Learning task-driven adaptive graphs may better serve GNN-based models for more accurate supervised single-cell classification.

To address these issues, we propose scBiGNN, a novel bilevel graph representation learning method that takes advantage of the above two kinds of biological networks in scRNA-seq classification. Specifically, scBiGNN consists of two GNN modules for cell type classification. A gene-level GNN based on the gene-gene interaction graph is designed to produce a representation for each individual cell, in which the interactions are adaptively learned via self-attention [23, 24]. Then the cell-cell graph is constructed by linking cells that have similar representations generated by the gene-level GNN (i.e., neighboring cells are more likely from the same class), upon which a cell-level GNN is built to improve cell representations by aggregating features from the neighborhood. Such a bilevel framework can integrate both gene- and cell-level structural information for more comprehensive scRNA-seq graph representation learning.

In real-world scRNA-seq datasets, the dimension of the gene expression data is commonly very high. Therefore, it is challenging to optimize our scBiGNN in an end-to-end manner when both the gene-gene and cell-cell graphs are large. Inspired by previous studies that optimize two neural networks in a whole framework [18, 35, 31], we employ the Expectation Maximization (EM) method to alternately train the two GNN modules via the E-step and M-step, which theoretically maximizes the evidence lower bound of the log-likelihood of the observed cell labels. In each step, one GNN is fixed to generate pseudo-labels for optimizing the other. In this alternating training fashion, scBiGNN can scale to large datasets, and gradually enhance the classification performance of both GNN modules by reinforcing each other. To validate the effectiveness of our proposed method, we conduct cell type classification on several popular benchmark datasets. Experimental results demonstrate that our scBiGNN achieves superior performance over a number of baselines.

## 2 Background

**Data Description and Pre-processing.** The scRNA-seq data can be represented as a gene expression matrix $\mathbf{X} \in \mathbb{R}^{N \times M}$, where $N$ is the number of cells, $M$ is the number of genes, and each element $X_{ij}$ indicates the expression counts of the $j$-th gene in the $i$-the cell. First, genes that have zero expression values across all the cells are removed. Then, the gene expression data of each cell is normalized

by $\tilde{X}_{ij} = \log\left(1 + s \cdot \frac{X_{ij}}{\sum_m X_{im}}\right)$ in which $s$ is a scaling value and is set as $10^6$ following existing work [32]. After normalization, the variance value of each gene across all the cells is calculated, and the top-$T$ variable genes are selected (e.g., $T = 1000$ [28]) while others are filtered out, based on which we can obtain the pre-processed gene expression data $\tilde{\mathbf{X}} \in \mathbb{R}^{N \times T}$.

**Graph Neural Networks (GNNs).** The goal of GNNs is to learn effective node and graph representations by iteratively aggregating information from the topological neighborhoods. Formally, let $\mathcal{G} = (\mathcal{V}, \mathbf{A})$ denote a graph where $\mathcal{V} = \{v_1, \ldots, v_n\}$ is the node set and $\mathbf{A} \in \mathbb{R}^{n \times n}$ is the adjacency matrix with each non-zero value $A_{ij}$ representing the edge between nodes $v_i$ and $v_j$. We also have a feature matrix $\mathbf{F} = [\mathbf{f}_1, \ldots, \mathbf{f}_n]^\top \in \mathbb{R}^{n \times f}$ with $\mathbf{f}_i \in \mathbb{R}^f$ being the feature vector of node $v_i$. The $l$-th feature aggregation layer of GNNs can take the form of $\mathbf{F}^{(l)} = \mathrm{MLP}^{(l)}\left(\mathrm{R}^{(l)}\left(\mathbf{A}\right)\mathbf{F}^{(l-1)}\mathbf{W}^{(l)}\right)$, in which $\mathbf{F}^{(l)} = [\mathbf{f}_1^{(l)}, \ldots, \mathbf{f}_n^{(l)}]^\top \in \mathbb{R}^{n \times f_l}$ is the matrix of node representations after $l$ aggregation layers, $\mathbf{F}^{(0)} = \mathbf{F}$, $\mathbf{W}^{(l)} \in \mathbb{R}^{f_{l-1} \times f_l}$ is a learnable weight matrix, $\mathrm{R}^{(l)}\left(\mathbf{A}\right)$ is some operator on the adjacency matrix such as normalization [11] and self-attention mechansim [24], and $\mathrm{MLP}^{(l)}\left(\cdot\right)$ is the multi-layer perception (MLP) for nonlinear feature transformation. After $L$ aggregation layers, we can obtain the node representations $\mathbf{F}^{(l)}$ ($l = 0, \ldots, L$). To produce a graph-level representation, we can employ a read-out function that pools all the node representations into a single vector $\mathbf{f}_{\mathcal{G}} = \mathrm{Read\text{-}Out}\left(\{\mathbf{F}_l\}_{l=0}^L\right)$, which is permutation invariant to the order of graph nodes, e.g., sum-pooling and max-pooling.

**GNN-based Single-cell Classification.** Existing GNN-based methods for cell type identification can be categorized into two groups. The first group leverages gene-gene relationships. Specifically, Yin et al. [32] collect several well-known gene interaction networks such as the STRING database [21], on which GNNs are applied to aggregate information from interacting genes for each expressed gene to improve cell representations. Wang et al. [28] also utilize the STRING database and propose sigGCN, in which a GNN-based autoencoder is used to reconstruct gene expression data and a fully connected neural network extracts features by taking the scRNA-seq data as input. The features generated by the encoder part of the autoencoder and the fully connected neural network are concatenated for cell type classification. HNNVAT [26] further introduces virtual adversarial training that adds noise to input data to make the model robust against noise perturbations. In addition, Yang et al. [30] propose scBERT, which follows BERT's pretraining and fine-tuning paradigm [10] with vast amounts of unlabelled scRNA-seq data and learns gene-gene interactions via Performers [6]. The second group of methods focuses on cell-level relationships. As more and more research demonstrates that cell-cell graph provides valuable structural information to learn effective cell representations for scRNA-seq analyses [19, 25, 5, 33, 34], Li et al. [13] benchmarks several GNNs for cell type classification based on the cell-cell graph and outperforms traditional machine learning methods. These results motivate us to combine the merits of these two types of methods to integrate gene- and cell-level structural information for more accurate cell type classification.

## 3 The Proposed Method: scBiGNN

In this section, we develop the scBiGNN framework for bilevel graph representation learning on scRNA-seq data, in which both gene- and cell-level relationships are exploited for more accurate cell type classification. We first introduce some preliminaries about the problem statement and the workflow of gene- and cell-level GNNs. Then, we overview our EM-based learning framework and elaborate the optimization procedures of the E-step and M-step respectively.

### 3.1 Preliminaries

**Problem Statement.** In our work, two biological networks are constructed based on the pre-processed gene expression data $\tilde{\mathbf{X}} \in \mathbb{R}^{N \times T}$. One is the gene-gene interaction graph $\mathcal{G}^{\mathrm{g}} = (\mathcal{V}^{\mathrm{g}}, \mathbf{A}^{\mathrm{g}})$ where $\mathcal{V}^{\mathrm{g}}$ is the set of $T$ genes $\{v_1^{\mathrm{g}}, \ldots, v_T^{\mathrm{g}}\}$ and $\mathbf{A}^{\mathrm{g}} \in \mathbb{R}^{T \times T}$ is the associated adjacency matrix representing the interactions between genes, the other is the cell-cell graph $\mathcal{G}^{\mathrm{c}} = (\mathcal{V}^{\mathrm{c}}, \mathbf{A}^{\mathrm{c}})$ where $\mathcal{V}^{\mathrm{c}} = \{v_1^{\mathrm{c}}, \ldots, v_N^{\mathrm{c}}\}$ indicates the set of $N$ cells in the scRNA-seq dataset and $\mathbf{A}^{\mathrm{c}} \in \mathbb{R}^{N \times N}$ is its adjacency matrix encoding the relationships between cells. Given $\tilde{\mathbf{X}}$ and the labels $\mathbf{Y}^L \in \mathbb{R}^{|\mathcal{V}^L| \times C}$ for a subset of annotated cells $\mathcal{V}^L \subset \mathcal{V}^{\mathrm{c}}$, the goal is to predict the labels $\mathbf{Y}^U \in \mathbb{R}^{|\mathcal{V}^U| \times C}$ for the the unlabeled cells $\mathcal{V}^U = \mathcal{V}^{\mathrm{c}} \backslash \mathcal{V}^L$, where $C$ is the number of cell types.

**Gene-level GNNs.** The gene-level GNNs aim to predict the label of each cell $v_i^c$ using its own gene expression data $\tilde{\mathbf{x}}_i$ (the $i$-th row of $\tilde{\mathbf{X}}$) and the gene-gene interaction network. Formally, they model the following label distribution $q_\phi(\mathbf{y}_i|\tilde{\mathbf{x}}_i)$ for each $v_i^c \in \mathcal{V}^c$ with parameters $\phi$:

$$\{\mathbf{f}_{i,t}\}_{t=1}^{T_i} = \text{GNN}^g\left(\tilde{\mathbf{x}}_i, \mathbf{A}^g\right),$$

$$\mathbf{h}_i = \text{Read-Out}\left(\{\mathbf{f}_{i,t}\}_{t=1}^{T_i}\right),$$

$$q_\phi(\mathbf{y}_i|\tilde{\mathbf{x}}_i) = \text{Cat}\left(\mathbf{y}_i\,\Big|\,\text{Clf}^g\left(\mathbf{h}_i\right)\right), \tag{1}$$

where $\text{GNN}^g(\cdot)$ denotes the node representation learning function of the gene-level GNN, $\{\mathbf{f}_{i,t}\}_{t=1}^{T_i}$ is the set of gene representations learned by $\text{GNN}^g(\cdot)$, $T_i$ is the number of genes that have non-zero expression values in $\tilde{\mathbf{x}}_i$, $\mathbf{h}_i$ is the representation of cell $v_i^c$, $\text{Clf}^g(\cdot)$ denotes a classifier, and $\text{Cat}(\cdot)$ stands for the categorical distribution. In this way, gene-gene structural information can be exploited from the local biological view of each individual cell to enhance cell representation learning.

**Cell-level GNNs.** The cell-level GNNs classify cell types by using the relationships among all the cells in the dataset. Specifically, the workflow can be characterized as below:

$$\{\mathbf{h}_i'\}_{i=1}^{N} = \text{GNN}^c\left(\{\tilde{\mathbf{x}}_i, \mathbf{h}_i\}_{i=1}^{N}, \mathbf{A}^c\right),$$

$$p_\theta(\mathbf{y}_i|\tilde{\mathbf{X}}) = \text{Cat}\left(\mathbf{y}_i\,\Big|\,\text{Clf}^c\left(\mathbf{h}_i'\right)\right), \tag{2}$$

where $\text{GNN}^c(\cdot)$ is the node representation learning function, $\{\mathbf{h}_i'\}_{i=1}^{N}$ is the set of cell representations learned by $\text{GNN}^c(\cdot)$, $\text{Clf}^c(\cdot)$ is a classifier, and $\theta$ is the model parameters. By aggregating features from neighboring cells with similar characteristics, cell-level GNNs extract the cell-cell structural information from a global view of the scRNA-seq dataset to improve single-cell classification.

Both gene- and cell-level GNNs have proved effective in scRNA-seq analyses and achieved superior performance for automatic cell type identification. Therefore, we propose scBiGNN, a bilevel graph representation learning method that aims to integrate these two levels of structural information to have a more comprehensive view of scRNA-seq data.

## 3.2 EM Framework

Directly cascading the above gene- and cell-level GNNs and training them end-to-end would face the scalability issue when the size of the scRNA-seq dataset is large. Moreover, as these two types of GNNs are complementary, end-to-end learning does not allow them to interact and enhance each other. Therefore, inspired by several previous studies [18, 35, 31], we instead employ the EM algorithm to train these two GNN modules alternately and make them gradually reinforce each other for more accurate cell type classification.

To be concrete, our proposed scBiGNN maximizes the evidence lower bound (ELBO) of the log-likelihood of the observed cell labels:

$$\log p_\theta(\mathbf{Y}^L|\tilde{\mathbf{X}}) \geq \mathbb{E}_{q_\phi(\mathbf{Y}^U|\tilde{\mathbf{X}})}\left[\log p_\theta(\mathbf{Y}^L, \mathbf{Y}^U|\tilde{\mathbf{X}}) - \log q_\phi(\mathbf{Y}^U|\tilde{\mathbf{X}})\right] \triangleq \mathcal{L}_{\text{ELBO}}(\theta, \phi; \mathbf{Y}^L, \tilde{\mathbf{X}}), \tag{3}$$

where $q_\phi(\mathbf{Y}^U|\tilde{\mathbf{X}})$ can be arbitrary distribution over $\mathbf{Y}^U$ (s.t. $q_\phi(\mathbf{Y}^U|\tilde{\mathbf{X}}) > 0$ if $p_\theta(\mathbf{Y}^L, \mathbf{Y}^U|\tilde{\mathbf{X}}) > 0$), and the equality holds when $q_\phi(\mathbf{Y}^U|\tilde{\mathbf{X}})$ equals to the true posterior distribution $p_\theta(\mathbf{Y}^U|\mathbf{Y}^L, \tilde{\mathbf{X}})$. As can be seen, Eq. (3) formalizes the objective function with two distributions $q_\phi$ and $p_\theta$, which can be modeled by the abovementioned gene- and cell-level GNNs respectively. According to the EM framework, we alternately train $q_\phi$ in the E-step (with $p_\theta$ fixed) and $p_\theta$ in the M-step (with $q_\phi$ fixed) to maximize $\mathcal{L}_{\text{ELBO}}(\theta, \phi; \mathbf{Y}^L, \tilde{\mathbf{X}})$. Thus, we can separately optimize these two GNN modules through several iterations, leading to better scalability and making them interact and enhance each other. In what follows, we introduce the structures of these two GNN modules and how they reinforce each other in the training procedure.

## 3.3 E-step

In the E-step, the cell-level GNN (i.e., $p_\theta$) is fixed and the gene-level GNN (i.e., $q_\phi$) is optimized to maximize $\mathcal{L}_{\text{ELBO}}(\theta, \phi; \mathbf{Y}^L, \tilde{\mathbf{X}})$. In this section, we first present the detailed structure of our employed gene-level GNN, and then we introduce the optimization of $q_\phi$ in the E-step.

**Structure of Gene-level GNN.** The input of the gene-level GNN module includes two embeddings. The first is gene embedding, which assigns a unique and learnable vector $\mathbf{e}_j^{\text{gene}} \in \mathbb{R}^{d_g}$ to each gene $v_j^{\text{g}}$. The second is count embedding, which transforms every non-zero gene expression value $\tilde{X}_{ij}$ of cell $v_i^{\text{c}}$ to a $d_g$-dimensional vector through an MLP, i.e., $\mathbf{e}_{ij}^{\text{count}} = \text{MLP}(\tilde{X}_{ij}) \in \mathbb{R}^{d_g}$. The input embedding of gene $v_j^{\text{g}}$ in cell $v_i^{\text{c}}$ is then defined as $\mathbf{e}_{ij} = \mathbf{e}_j^{\text{gene}} + \mathbf{e}_{ij}^{\text{count}}$. By gathering and stacking all the input embeddings of genes that have non-zero expression values in cell $v_i^{\text{c}}$, we have the input embedding matrix $\mathbf{E}_i \in \mathbb{R}^{T_i \times d_g}$ for cell $v_i^{\text{c}}$.

Inspired by attention-based models [23, 24], the gene-level GNN of scBiGNN adaptively learns gene-gene interactions via the self-attention mechanism. Specifically, the feature aggregation layer of $\text{GNN}^{\text{g}}(\cdot)$ is defined as

$$\text{R}^{(l)}\left(\mathbf{A}^{\text{g}}\right) = \text{softmax}\left((\mathbf{F}_i^{(l-1)}\mathbf{W}_{\text{g}}^{(l)})(\mathbf{F}_i^{(l-1)}\mathbf{W}_{\text{g}}^{(l)})^\top\right),$$

$$\mathbf{F}_i^{(l)} = \text{MLP}_{\text{g}}^{(l)}\left(\text{R}^{(l)}\left(\mathbf{A}^{\text{g}}\right)\mathbf{F}_i^{(l-1)}\mathbf{W}_{\text{g}}^{(l)}\right), \tag{4}$$

where $\text{softmax}(\cdot)$ is the row-wise softmax operation, $\mathbf{W}_{\text{g}}^{(l)}$ is a learnable weight matrix, $\text{MLP}_{\text{g}}^{(l)}(\cdot)$ is an MLP, $\mathbf{F}_i^{(l)}$ is the output feature matrix of the $l$-th layer, and $\mathbf{F}_i^{(0)} = \mathbf{E}_i$ for each cell $v_i^{\text{c}}$. Moreover, in each layer of $\text{GNN}^{\text{g}}(\cdot)$, we can also employ the multi-head attention mechanism [23], which performs several different attention functions of Eq. (4) and concatenates their outputs as the feature matrix. After $L$ layers, we can obtain the set of gene representations $\{\mathbf{f}_{i,t}^j\}_{t=1}^{T_i}$ in which $\mathbf{f}_{i,t}^j$ is the $t$-th row of $\mathbf{F}_i^{(L)}$ and the superscript $j$ indicates that it corresponds to the gene $v_j^{\text{g}}$. The read-out function in Eq. (1) can take the form of the weighted sum of all the gene representations to produce the cell representation for $v_i^{\text{c}}$:

$$\mathbf{h}_i = \text{Read-Out}\left(\{\mathbf{f}_{i,t}^j\}_{t=1}^{T_i}\right) = \sum_j \alpha_j \mathbf{f}_{i,t}^j, \tag{5}$$

where $\alpha_j$ can be a learnable weight for gene $v_j^{\text{g}}$ ($j = 1, \ldots, T$) or set as $1/T_i$ which performs average pooling. Then, $q_\phi(\mathbf{y}_i|\tilde{\mathbf{x}}_i)$ can be obtained by feeding $\mathbf{h}_i$ to an MLP classifier, as shown in Eq. (1).

**Optimization of Gene-level GNN.** Maximizing $\mathcal{L}_{\text{ELBO}}(\theta, \phi; \mathbf{Y}^L, \tilde{\mathbf{X}})$ in the E-step is equivalent to making $q_\phi(\mathbf{Y}^U|\tilde{\mathbf{X}})$ approximate the true posterior distribution $p_\theta(\mathbf{Y}^U|\mathbf{Y}^L, \tilde{\mathbf{X}})$. According to the EM framework, the standard operation is to minimize the Kullback-Leibler (KL) divergence $\text{KL}(q_\phi(\mathbf{Y}^U|\tilde{\mathbf{X}})\|p_\theta(\mathbf{Y}^U|\mathbf{Y}^L, \tilde{\mathbf{X}}))$. However, directly optimizing the KL divergence is nontrivial, as it depends on the entropy of $q_\phi(\mathbf{Y}^U|\tilde{\mathbf{X}})$, which is hard to deal with [35]. Moreover, for $v_i^{\text{c}} \in \mathcal{V}^U$, when $p_\theta(\mathbf{y}_i|\mathbf{Y}^L, \tilde{\mathbf{X}})$ is close to the one-hot categorical distribution, it would face the instability issue caused by $\log y$ with $y$ approaching zero. Therefore, following several EM-based optimization methods [18, 35, 31], we instead treat the fixed $p_\theta(\mathbf{Y}^U|\mathbf{Y}^L, \tilde{\mathbf{X}})$ as the target and minimize the reverse KL divergence $\text{KL}(p_\theta(\mathbf{Y}^U|\mathbf{Y}^L, \tilde{\mathbf{X}})\|q_\phi(\mathbf{Y}^U|\tilde{\mathbf{X}}))$ to make $q_\phi(\mathbf{Y}^U|\tilde{\mathbf{X}})$ approximate the target. Assuming that the labels of different cells are independent, which is reasonable as the organization and function of each cell are not determined by other cells, we can derive the following loss function for unlabeled cells:

$$\mathcal{L}_E^U = -\mathbb{E}_{p_\theta(\mathbf{Y}^U|\mathbf{Y}^L, \tilde{\mathbf{X}})}\left[\log q_\phi(\mathbf{Y}^U|\tilde{\mathbf{X}})\right] = -\sum_{v_i^{\text{c}} \in \mathcal{V}^U} \mathbb{E}_{p_\theta(\mathbf{y}_i|\mathbf{Y}^L, \tilde{\mathbf{X}})}\left[\log q_\phi(\mathbf{y}_i|\tilde{\mathbf{x}}_i)\right], \tag{6}$$

where $p_\theta(\mathbf{y}_i|\mathbf{Y}^L, \tilde{\mathbf{X}}) = p_\theta(\mathbf{y}_i|\tilde{\mathbf{X}})$ is the categorical distribution predicted by the cell-level GNN for $v_i^{\text{c}} \in \mathcal{V}^U$ in the previous M-step. That is, the pseudo-labels predicted by the cell-level GNN are used for training the gene-level GNN. Additionally, the gene-level GNN can also be trained by the ground-truth labels of cells in $\mathcal{V}^L$. Therefore, we have the loss function for labeled cells:

$$\mathcal{L}_E^L = -\sum_{v_i^{\text{c}} \in \mathcal{V}^L} \log q_\phi(\mathbf{y}_i|\tilde{\mathbf{x}}_i), \tag{7}$$

where $\mathbf{y}_i$ is the ground-truth label for cell $v_i^{\text{c}} \in \mathcal{V}^L$. Combining Eqs. (6) and (7), the overall objective in the E-step for optimizing $\phi$ is to minimize $\mathcal{L}_E = \mathcal{L}_E^U + \beta\mathcal{L}_E^L$ with $\beta$ being a balancing hyper-parameter (we set $\beta = 1$ in this work), which can be solved via stochastic gradient descent (SGD). To be more specific, in each update step of SGD, we sample a batch of labeled cells $\mathcal{B}^L$ and a batch of unlabeled cells $\mathcal{B}^U$ to perform gradient descent $\phi \leftarrow \phi - \gamma_{\text{g}}\nabla_\phi\tilde{\mathcal{L}}_E$, where $\tilde{\mathcal{L}}_E = -\sum_{v_i^{\text{c}} \in \mathcal{B}^U} \log q_\phi(\hat{\mathbf{y}}_i|\tilde{\mathbf{x}}_i) - \beta\sum_{v_i^{\text{c}} \in \mathcal{B}^L} \log q_\phi(\mathbf{y}_i|\tilde{\mathbf{x}}_i)$ with $\hat{\mathbf{y}}_i$ sampled by $\hat{\mathbf{y}}_i \sim p_\theta(\mathbf{y}_i|\tilde{\mathbf{X}})$ for unlabeled cells and $\gamma_{\text{g}}$ is the learning rate.

### 3.4 M-step

In the M-step, the gene-level GNN (i.e., $q_\phi$) is fixed and the cell-level GNN (i.e., $p_\theta$) is updated to maximize $\mathcal{L}_{\text{ELBO}}(\theta, \phi; \mathbf{Y}^L, \tilde{\mathbf{X}})$. Also, before we introduce the optimization of $p_\theta$, we present the detailed workflow of our cell-level GNN.

**Structure of Cell-level GNN.** Existing cell-level GNNs predetermine the cell-cell graph based on $\tilde{\mathbf{X}}$ [13]. By contrast, we construct the cell-cell graph by performing $k$-nearest neighbors analysis on the cell representations $\{\mathbf{h}_i\}_{i=1}^N$ generated by the gene-level GNN in each EM iteration, i.e., $\mathbf{A}^c = \text{kNN-graph}(\{\mathbf{h}_i\}_{i=1}^N)$, since $\{\mathbf{h}_i\}_{i=1}^N$ should have better clustering characteristics than $\tilde{\mathbf{X}}$ for identifying cell types and its quality is enhanced during the optimization of $q_\phi$.

Given the gene expression data $\tilde{\mathbf{x}}_i$ and the cell representation $\mathbf{h}_i$, the input feature for each cell $v_i^c$ is defined as $\mathbf{z}_i = \text{Concat}(\mathbf{h}_i, \text{MLP}(\tilde{\mathbf{x}}_i)) \in \mathbb{R}^{d_c}$, where $\text{Concat}(\cdot)$ denotes the concatenation operation for the input vectors. By stacking the input features of all the cells, we have the input feature matrix $\mathbf{Z} \in \mathbb{R}^{N \times d_c}$. And the structure of $\text{GNN}^c(\cdot)$ can be modeled as:

$$\text{R}^{(l)}(\mathbf{A}^c) = \mathbf{D}^{-1}\mathbf{A}^c, \quad \mathbf{D} = \text{diag}(\mathbf{A}^c \mathbf{1}_N),$$
$$\mathbf{Z}^{(l)} = \text{MLP}_c^{(l)}\left(\text{R}^{(l)}(\mathbf{A}^c)\mathbf{Z}^{(l-1)}\mathbf{W}_c^{(l)}\right), \tag{8}$$

where $\mathbf{1}_N$ is the all-ones vector of length $N$, $\text{diag}(\cdot)$ returns a diagonal matrix with the input vector as the diagonal, $\mathbf{W}_c^{(l)}$ is a learnable weight matrix, $\text{MLP}_c^{(l)}(\cdot)$ is an MLP, $\mathbf{Z}^{(l)}$ is the output feature matrix of the $l$-th layer, and $\mathbf{Z}^{(0)} = \mathbf{Z}$. As for $\mathbf{h}_i'$ that is fed to an MLP classifier in Eq. (2), we find that setting $\mathbf{h}_i' = \text{Concat}(\mathbf{z}_i^{(0)}, \ldots, \mathbf{z}_i^{(L)})$ with $\mathbf{z}_i^{(l)}$ being the $i$-th row of $\mathbf{Z}^{(l)}$ practically achieves better classification performance for $p_\theta(\mathbf{y}_i|\tilde{\mathbf{X}})$, which is similar to the jumping knowledge networks [29].

**Optimization of Cell-level GNN.** Maximizing $\mathcal{L}_{\text{ELBO}}(\theta, \phi; \mathbf{Y}^L, \tilde{\mathbf{X}})$ in the M-step is equivalent to minimizing the following loss function:

$$\mathcal{L}_M = -\mathbb{E}_{q_\phi(\mathbf{Y}^U|\tilde{\mathbf{X}})}\left[\log p_\theta(\mathbf{Y}^L, \mathbf{Y}^U|\tilde{\mathbf{X}})\right]$$
$$= -\sum_{v_i^c \in \mathcal{V}^U} \mathbb{E}_{q_\phi(\mathbf{y}_i|\tilde{\mathbf{x}}_i)}\left[\log p_\theta(\mathbf{y}_i|\tilde{\mathbf{X}})\right] - \sum_{v_i^c \in \mathcal{V}^L} \log p_\theta(\mathbf{y}_i|\tilde{\mathbf{X}}). \tag{9}$$

Here, the first term is the loss for unlabeled cells, which trains $p_\theta$ using the pseudo-labels generated by the gene-level GNN in the previous E-step. The second term is the loss for labeled cells, which is the standard supervised learning with ground-truth labels. Minimizing Eq. (9) w.r.t. $p_\theta$ can also be solved via SGD. Concretely, in each update step of SGD, we sample $\hat{\mathbf{y}}_i \sim q_\phi(\mathbf{y}_i|\tilde{\mathbf{x}}_i)$ for unlabeled cells and perform gradient descent $\theta \leftarrow \theta - \gamma_c \nabla_\theta \tilde{\mathcal{L}}_M$, in which $\tilde{\mathcal{L}}_M = -\sum_{v_i^c \in \mathcal{V}^U} \log p_\theta(\hat{\mathbf{y}}_i|\tilde{\mathbf{X}}) - \sum_{v_i^c \in \mathcal{V}^L} \log p_\theta(\mathbf{y}_i|\tilde{\mathbf{X}})$ and $\gamma_c$ is the learning rate.

### 3.5 Overall Optimization

To optimize the overall scBiGNN framework, we first pre-train the gene-level GNN with labeled cells to obtain the initial $q_\phi$. Then, the EM algorithm alternately optimizes $p_\theta$ (i.e., the cell-level GNN) and $q_\phi$ (i.e., the gene-level GNN) to reinforce each other. In the M-step, the cell representations generated by the gene-level GNN are used to construct the cell-cell graph, and the pseudo-labels predicted by $q_\phi$ together with the given cell labels are utilized to train the cell-level GNN. In the E-step, the pseudo-labels produced by $p_\theta$ and the ground-truth cell labels are used to train the gene-level GNN, in which the gene-gene interactions are adaptively learned. After several EM iterations, the performance of both cell- and gene-level GNNs is enhanced in cell type classification.

## 4 Experiments

### 4.1 Benchmark Datasets

In this work, we conduct cell type classification experiments on five benchmark datasets to evaluate the performance of our scBiGNN, i.e., Zheng68K, Zhengsorted, BaronHuman, BaronMouse and AMB, which are widely used in many published papers for single-cell classification. These datasets

Table 1: Statistics of the five scRNA-seq benchmark datasets.

| Dataset | Description | Protocol | Species | #Genes | #Cells | #Cell types |
|---|---|---|---|---|---|---|
| Zheng68K | PBMC | 10X Chromium | Homo sapiens | 20387 | 65943 | 11 |
| Zhengsorted | FACS-sorted PBMC | 10X Chromium | Homo sapiens | 21952 | 20000 | 10 |
| BaronHuman | Human pancreas | inDrop | Homo sapiens | 17499 | 8569 | 14 |
| BaronMouse | Mouse pancreas | inDrop | Mus musculus | 14861 | 1886 | 13 |
| AMB | Primary mouse visual cortex | SMART-Seq v4 | Mus musculus | 42625 | 12832 | 22 |

Table 2: Classification accuracy of all the methods on the five datasets.

| Method | Zheng68K | Zhengsorted | BaronHuman | BaronMouse | AMB |
|---|---|---|---|---|---|
| SVM | 0.701 | 0.829 | 0.978 | 0.973 | 0.992 |
| LDA | 0.662 | 0.692 | 0.978 | 0.952 | 0.901 |
| NMC | 0.597 | 0.724 | 0.912 | 0.919 | 0.976 |
| RF | 0.674 | 0.796 | 0.962 | 0.968 | 0.985 |
| scID | 0.525 | 0.743 | 0.535 | 0.338 | 0.906 |
| CHETAH | 0.298 | 0.645 | 0.925 | 0.895 | 0.939 |
| SingleR | 0.673 | 0.718 | 0.968 | 0.908 | 0.962 |
| ACTINN | 0.724 | 0.825 | 0.977 | 0.978 | 0.992 |
| GCN-C | 0.710 | 0.831 | 0.978 | 0.977 | 0.992 |
| scGraph | 0.729 | 0.835 | 0.983 | 0.976 | 0.991 |
| sigGCN | 0.733 | 0.844 | 0.979 | 0.978 | 0.992 |
| HNNVAT | 0.734 | 0.846 | 0.982 | 0.979 | 0.992 |
| **scBiGNN** ($q_\phi$) | 0.757 | 0.860 | 0.981 | 0.979 | 0.993 |
| **scBiGNN** ($p_\theta$) | **0.760** | **0.867** | **0.983** | **0.983** | **0.994** |

can be directly downloaded from Zenodo (`https://doi.org/10.5281/zenodo.3357167`), and the detailed information is depicted in Table 1.

## 4.2 Implementations

As for the gene-level GNN module, we use a one-layer model with four attention heads for the BaronMouse dataset, while a two-layer model with a single head in each layer is employed for the other datasets. Additionally, for Zheng68K and Zhengsorted datasets, the read-out function is set as the simple average pooling operation, while for the others we find that using learnable $\alpha_j$ in Eq. (5) performs better. As for the cell-level GNN module, a three-layer model is employed for cell graph representation learning. The feature dimension of each layer's output is set as 32 for both GNN modules. All the MLPs used in scBiGNN have one hidden layer with 32 neurons and ReLU

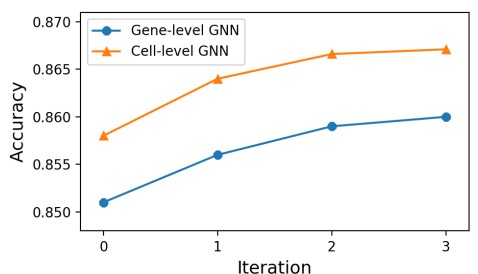

Figure 1: The convergence curves of $q_\phi$ and $p_\theta$ on the Zhengsorted dataset.

activation. The maximum number of EM iterations is set as 3, which is commonly enough for scBiGNN to converge, as shown in Figure 1.

## 4.3 Comparison Methods

We compare with a variety of baseline models to demonstrate the superior performance of scBiGNN. Specifically, following the previous study [32], eight methods are considered: support vector machine (SVM), linear discriminant analysis (LDA), nearest mean classifier (NMC), random forest (RF), scID [4], CHETAH [7], SingleR [3] and ACTINN [15]. In addition, three recent works that use gene-level GNNs are included: scGraph [32], sigGCN [28] and HNNVAT [26]. We also compare with the cell-level GNN, termed GCN-C, in which the cell-cell graph is constructed by principal component analysis and $k$-nearest neighbors analysis on the raw gene expression data [13].

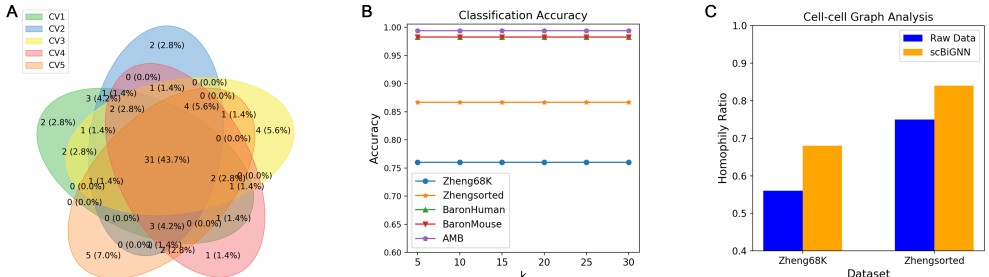

Figure 2: Network analysis of gene-gene and cell-cell graphs. (A) Venn plot of five important gene lists extracted from gene-gene interactions with five-fold cross validation on the BaronMouse dataset. (B) Ablation study on $k$ of cell-cell $k$NN graphs. (C) Homophily analysis on the cell-cell graphs.

## 4.4 Classification Results

We benchmark scBiGNN against all the baselines by performing five-fold cross validation on each dataset. The results of cell type classification accuracy are summarized in Table 2, in which the best performance in each column is highlighted in bold. As can be seen, among all the baselines, graph-based deep learning methods are the most accurate classifiers, verifying that the biological structural information is beneficial to identifying cell types from scRNA-seq data. Out of all the graph-based models, our scBiGNN consistently achieves the highest accuracy on all the benchmark datasets. Noticeably, it outperforms all the baselines with more than 2.5% accuracy improvement on the largest scRNA-seq dataset (i.e., Zheng68K). Moreover, from Table 2 and Figure 1 we can observe that $p_\theta$ always performs better than $q_\phi$. The reason is that $p_\theta$ explicitly integrates the bilevel graph representations of $\mathbf{h}_i$ that summarizes gene-level structural information and $\mathbf{A}^c$ that contains cell-level structural information. Overall, these results validate that the gene- and cell-level relationships are effectively mined by $p_\theta$ of scBiGNN to facilitate scRNA-seq classification.

## 4.5 Network Analysis

**Gene-gene Graph.** Our scBiGNN adaptively learns gene-gene interactions by the self-attention function $\mathrm{R}^{(l)}\left(\mathbf{A}^g\right)$ in Eq. (4). We integrate all the attention matrices into a gene-gene interaction matrix $\tilde{\mathbf{A}}$ by taking the average across all the corresponding elements in all attention matrices for every gene-gene pair, where each element $\tilde{A}_{ij}$ measures the attention of gene $v_i^g$ paid to gene $v_j^g$. Then, we can obtain the importance score $s_j$ for each gene $v_j^g$, which is defined as $s_j = \sum_i \tilde{A}_{ij}$. We sort the importance scores and get five lists of top 50 important genes from five-fold cross validation. Figure 2(A) shows that most genes appear more than once in the five lists, indicating that scBiGNN is trustworthy and effective in learning consistent important genes across different training procedures.

**Cell-cell Graph.** The cell-cell graph in scBiGNN is constructed as the $k$-nearest neighbor ($k$NN) graph based on the cell representations learned from $q_\phi$. In Figure 2(B), we can observe that our model is insensitive to $k$. A small value of $k$ (e.g., $k=5$) is enough for $p_\theta$ to achieve good performance, verifying the high quality of cell representations generated by $q_\phi$ in finding close neighborhoods. In Figure 2(C), we measure the homophily ratio (i.e., the proportion of intra-class edges in all the edges) of cell-cell graphs constructed by scBiGNN and the raw data $\tilde{\mathbf{X}}$ [13] respectively. As can be seen, cell-cell graphs built by scBiGNN link more cells of the same type, which is critical to learning separable cell representations across different classes to ease GNN-based node classification [31].

## 5 Conclusion

In this paper, we propose a novel bilevel graph representation learning framework termed scBiGNN to improve GNN-based cell type classification in scRNA-seq analysis. scBiGNN consists of two GNN modules, namely gene- and cell-level GNNs, to adaptively extract structural information at both biological levels. The EM algorithm is employed to optimize the two GNN modules by alternating between the E-step and M-step. In each step, the pseudo-labels predicted by one module are used to train the other, making them gradually reinforce each other. Experiments on five scRNA-seq datasets verify the effectiveness of scBiGNN compared to competing cell type classification methods.

## Acknowledgments and Disclosure of Funding

This work was supported in part by the National Natural Science Foundation of China under Grant 62250055, Grant 62371288, Grant 62320106003, Grant 61932022, Grant 61971285, Grant 62120106007, and in part by the Program of Shanghai Science and Technology Innovation Project under Grant 20511100100.

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
