# OpenReview forum: "scBiGNN: Bilevel Graph Representation Learning for Cell Type Classification from Single-cell RNA Sequencing Data"
_NeurIPS.cc/2023/Workshop/AI4Science — NeurIPS2023-AI4Science Poster_

### Official Review · Reviewer_KdkQ · 2023-10-25
**Review of scBiGNN**

**Rating:** 6
**Confidence:** 3

**Review:**

__Summary__:
The authors develop scBiGNN, a cell typing method that infers single-cell transcriptomic embeddings with a gene-level GNN and uses these embeddings to construct a graph of all cells in the dataset, which can then be used for robust cell-type classification.

__Strengths__:
- The method is elegant and well-formulated. Biological signals are inherently noisy and that including a contextual neighborhood of cells with similar latent states improves representation robustness for downstream tasks. This is justified by Figure 1.
- There are comparisons on several canonical datasets indicating strong performance.
- The results are easy to read and well presented. The manuscript overall is well written.

__Weaknesses__:
- The baselines are not comprehensive and many state of the art methods are missing from the results comparison. Seurat is the most alarming omission, but other prominent methods including scVI, SciBet, CellID, scGPT, and scBERT (which was mentioned in related work) are missing, and reported results from some of these works seem to exceed results reported for scBiGNN.

__Nits__:
- Is it necessary to transform scalar gene expression values into a d-dimensional embedding?

__Recommendation__:
The authors apply biological intuition to develop a novel method that improves performance on the canonical cell typing task. Empirical appear strong, but are undermined by notable baseline omissions. I tentatively recommend acceptance for the paper in its current state based on the method and preliminary results, but the results are questionable without better baseline comparisons.

---

### Meta-Review · Area_Chair_Ssxy · 2023-10-27

**Recommendation:** Accept (Poster)
**Confidence:** 5

**Metareview:**

This paper presents the Graph Neural Networks (GNNs) based model for cell type classification tasks on single-cell RNA-seq (scRNA-seq )data. The author(s) proposes that utilizing both cell-cell and gene-gene relationships is essential and can improve complementary by making them gradually reinforce each other for more accurate cell type classification. This paper offers an intriguing perspective by highlighting the benefits and biological significance of complementary training in both Cell-level GNN and Gene-level GNN.

However, as the reviewer pointed out, it is crucial to include comparisons with state-of-the-art baselines to establish the effectiveness of the proposed method. Furthermore, conducting more rigorous ablation studies, specifically investigating the utilization of only cell-level or gene-level graphs, is essential to substantiate the advantages of leveraging both perspectives.